# The Pathogenic RET Val804Met Variant in Acromegaly: A New Clinical Phenotype?

**DOI:** 10.3390/ijms25031895

**Published:** 2024-02-05

**Authors:** Sabrina Chiloiro, Ettore Domenico Capoluongo, Flavia Costanza, Angelo Minucci, Antonella Giampietro, Amato Infante, Domenico Milardi, Claudio Ricciardi Tenore, Maria De Bonis, Simona Gaudino, Guido Rindi, Alessandro Olivi, Laura De Marinis, Alfredo Pontecorvi, Francesco Doglietto, Antonio Bianchi

**Affiliations:** 1Department of Translational Medicine and Surgery, Università Cattolica del Sacro, 00168 Rome, Italy; flavia.costanza.fc@gmail.com (F.C.); antonella.giampietro@tiscali.it (A.G.); domenico.milardi@policlinicogemelli.it (D.M.); laurademarinis@yahoo.it (L.D.M.); ipofisipoliclinicogemelli@gmail.com (A.P.); abianchi68@yahoo.it (A.B.); 2Division of Endocrinology and Metabolism, Fondazione Policlinico Universitario A. Gemelli IRCCS, 00168 Rome, Italy; 3Department of Molecular Medicine and Medical Biotechnology, Federico II University, 80138 Naples, Italy; edotto70@gmail.com; 4Department of Clinical Pathology and Genomics, Ospedale per l’Emergenza Cannizzaro, 95126 Catania, Italy; 5Unit of Molecular Diagnostics and Genomics, Department of Laboratory Sciences and Infectious Diseases, Fondazione Policlinico Universitario A. Gemelli IRCCS, 00168 Rome, Italy; angelo.minucci@policlinicogemelli.it (A.M.); maria.debonis@policlinicogemelli.it (M.D.B.); 6Department of Imaging, Radiation Therapy and Hematology, Università Cattolica del Sacro Cuore, Fondazione Policlinico Universitario Agostino Gemelli, Istituto di Ricovero e Cura a Carattere Scientifico (IRCCS), 00168 Rome, Italy; amato.infante@policlinicogemelli.it (A.I.); simona.gaudino@policlinicogemelli.it (S.G.); 7Section of Anatomic Pathology, Department of Life Sciences and Public Health, Università Cattolica del Sacro Cuore, 00168 Rome, Italy; guido.rindi@policlinicogemelli.it; 8Unit of Head and Neck, Thoracic and Endocrine Pathology, Department of Woman and Child Health and Public Health, Fondazione Policlinico Universitario A. Gemelli IRCCS, Roma European Neuro-Endocrine Tumor Society (ENETS) Center of Excellence, 00168 Rome, Italy; 9Department of Neurosurgery, Fondazione Policlinico Universitario A. Gemelli IRCCS, Università Cattolica del Sacro Cuore, L.go A. Gemelli, 8, 00168 Rome, Italy; alessandro.olivi@policlinicogemelli.it (A.O.); francesco.doglietto@policlinicogemelli.it (F.D.)

**Keywords:** RET mutation, acromegaly, genetics, precision medicine, hereditary cancer-predisposing syndrome

## Abstract

Several genetic investigations were conducted to identify germline and somatic mutations in somatotropinomas, a subtype of pituitary tumors. To our knowledge, we report the first acromegaly patient carrying a *RET* pathogenic variant: c.2410G>A (rs79658334), p.Val804Met. Alongside the fact that the patient’s father and daughter carried the same variant, we investigated the clinical significance of this variant in the context of somatotropinomas and other endocrine tumors, reviewing the *RET* mutations’ oncogenic mechanisms. The aim was to search for new targets to precisely manage and treat acromegaly. Our case describes a new phenotype associated with the *RET* pathogenic variant, represented by aggressive acromegaly, and suggests consideration for *RET* mutation screening if NGS for well-established PitNET-associated gene mutations renders negative.

## 1. Introduction

Genetic discoveries have improved our understanding of the etiology and pathogenesis of several diseases [1]. This is also true for acromegaly, an endocrine disorder secondary to the hypersecretion of growth hormone (GH, or somatotroph hormone) and insulin-like growth factor 1 (IGF-I). Acromegaly is usually due to pituitary neuroendocrine tumors (PitNETs) such as somatotropinomas, mixed PitNETs such as mammo-somatotropinomas, or plurihormonal PitNETs [2,3].

Germline and somatic mutations have been extensively investigated in acromegalic patients [4,5]. Somatic mutations were identified in 40% of cases, and germline mutations were identified in 5% of cases [6]. Germline mutations may affect different genes, such as *AIP* (Aryl Hydrocarbon Receptor Interacting Protein), *MEN1* (Menin 1), *CDKN1B* (Cyclin Dependent Kinase Inhibitor 1B), *GPR101* (G Protein-Coupled Receptor 101), *PRKAR1A* (Protein Kinase CAMP-Dependent Type I Regulatory Subunit Alpha), and *GNAS* (GNAS Complex Locus) [7]. This, in turn, could impact our therapeutic approach and prognosis.

The germline mutation of the *AIP* gene (11q13.3) has a 3.6% prevalence in sporadic tumors, 4% in sporadic acromegaly, 29% in gigantism, and 10% in familial isolated pituitary adenoma (FIPA) syndrome [7,8,9,10]. *AIP*-mutated patients typically develop isolated, large, and invasive PitNETs that are resistant to conventional treatments. Histological studies often show sparsely granulated cytokeratin tumor patterns with absent or low expression of the somatostatin receptor 2 (SSTR2) [7,8,11,12].

The *MEN1* gene (11q13.1) and germline pathogenic variants (PVs) cause the Multiple Endocrine Neoplasia Type 1 (MEN-1) syndrome. MEN1 mutations have a prevalence of 0.6–2.6% of PitNETs [7,8,13,14]. The *CDKN1B* gene (12p13.1) germline PVs have been recently associated with MEN-4 syndrome, which is characterized by pituitary hyperplasia or tumor, primary hyperparathyroidism, and, rarely, pancreatic neuroendocrine neoplasia [7,8,15].

X-linked acro-gigantism (X-LAG) was described as a syndrome due to either germline or somatic duplications of the *GPR101* gene (Xq26.3) [3,4,11]. After *AIP* mutations, X-LAG is the second most common genetic cause of gigantism, usually occurring in the first years of life. It is often associated with large somatomammotropinoma or pituitary hyperplasia [7,8,16,17].

The *PRKAR1A* gene (17q22-24) germline PVs are solely described in acromegaly or gigantism, with 1% prevalence [7,8,18]. Most patients carry somatomammotropinoma or pituitary hyperplasia and, phenotypically, show the Carney complex phenotype [7,8,19].

The mosaic *GNAS* mutation (20q13.3) is rare and associated with McCune-Albright syndrome [7,8,20], while somatic mutations in *GNAS* are present in up to 40% of pituitary tumors [21]. *GNAS*-mutated somatotropinomas are less likely to be invasive but are characterized by higher GH and IGF-I secretion. Compared to *AIP*-mutated pituitary adenomas, *GNAS*-positive tumors arise in older patients and respond better to first-generation somatostatin ligands (SRLs). Furthermore, a higher expression of dopamine receptor 2 has been observed, so *GNAS* status could predict a better response to dopamine agonists (DA) [7,8,22].

Although genetic tests have clarified the origin of hereditary pituitary tumors, there are still some cases of acromegaly whose gene alterations have not yet been identified. To date, *RET* mutations have not been reported in acromegaly, if not exclusively in the context of the *MEN*. 

To our knowledge, we are describing the first acromegaly patient carrying the *RET* pathogenic variant c.2410G>A; p.Val804Met; rs79658334.

## 2. Case Description

A 48-year-old female patient was referred to our Pituitary Unit in July 2004 for a progressive change in facial appearance and the enlargement of her hands and feet. Hormonal assays documented: PRL 141.7 ng/mL (normal range: 3.5–15.5), IGF-I: 970 ng/mL (normal range: 115–307), basal GH: 70 ng/mL, and GH-nadir during oral glucose tolerance test: 35.7 ng/mL. Pituitary magnetic resonance imaging (MRI) revealed a giant pituitary tumor (maximum diameter 48 mm), invading both cavernous sinuses and the clivus (Figure 1a,b). In August 2004, the patient underwent trans-sphenoidal tumor removal. The histological examination showed positive immunohistochemistry for GH and prolactin, with a proliferative index (Ki67) of 1.5%. Due to the subtotal tumor removal, the co-secretion of prolactin, as well as the persistence of elevated GH (22.1 ng/mL), IGF-I (897 ng/mL, range: 115–307), and PRL levels (75 ng/mL), the patient received medical therapy with first-generation Somatostatin Receptor Ligand (fg-SRL) (Lanreotide 120 mg monthly) and DA (Cabergoline 0.25 mg/weekly). During the follow-up, the treatment doses were adjusted due to the persistence of high GH and IGF-I levels. The annual MRIs showed a stable residual. 

In 2012, the patient was diagnosed with a right breast fibro-adenoma and a left breast moderately differentiated invasive duct adenocarcinoma. The patient underwent a left supero-external quadrantectomy with lymphadenectomy and subsequent chemoradiotherapy. 

Due to the somatotropinoma’s aggressive behavior and the patient’s cancer history, a virtual panel was designed for PitNETs via clinical exome sequencing (CES) encompassing the following genes: *ABCC*, *ABCC8*, *AIP*, *APC*, *ATP2B3*, *CACNA1D*, *CACNA1H*, *CDKN1B*, *CLCN2*, *DICER1*, *DIS3L2*, *FGFR3*, *GCK*, *GIPR*, *GNAS*, *HNF1A*, *HNF4A*, *HRAS*, *IGF1R*, *KCNJ5*, *LHX3*, *MEN1*, *NF1*, *NSD1*, *PDE11A*, *PDE8B*, *POU1F*, *PRKAR1A*, *PROP1*, *PTCH1*, *PTEN*, *RET*, *SDHA*, *SDHB*, *SDHC*, *SDHD*, *SHANK3*, *SLC16A1*, *SLC34A1*, *TSC1*, *TSC2*, *VHL*. CES was performed using SOPHiA Clinical Exome Solution, covering the coding regions (±5 bp of intronic regions) of 4490 genes. Sequencing was performed on the MiSeq platform (Illumina MiSeq^®^ v3; 2 × 300 bp), with coverage of 50×. The NGS pipeline analysis results were analyzed using the SOPHiA DDM software (version v5.10.45). 

After an accurate variant calling process, the only pathogenic variant reported by the SoPHiA DDM software was the c.2410G>A (rs79658334), p.Val804Met variant in the RET gene. We further extended our analysis to the other genes outside the virtual panel, and no pathogenic variants were identified. To clarify the meaning of this mutation and its function, the genetic research portals ClinVar, Varsome, Franklin Uniprot, Ensembl, OMIM, Gene Cards, and gnomAD were consulted, and unanimously, this variant was classified as pathogenic (Class 5). These website portals reported that this pathogenic variant (PV) is associated with MEN-2A and MEN-2B, familial medullary thyroid carcinoma (FMTC), pheochromocytoma, hereditary cancer-predisposing syndrome (HCPS), renal hypodysplasia or aplasia, congenital central hypoventilation syndrome (CCHS), and Hirschsprung disease. 

Furthermore, we screened the patients for medullary thyroid cancer, pheochromocytoma, and paraganglioma. We also found serum calcitonin to be undetectable. A thyroid ultrasound showed a hypo-echogenic nodule sited on the right; the cytology examination excluded thyroid malignant neoplasia (Figure 1c). The 24 h metanephrine and catecholamine levels were within normal ranges. 

A total body CT and 18F-fluorodeoxyglucose positron emission tomography (18F-FDG-PET)/CT ruled out adrenal nodules, pheochromocytoma, or other neoplasia sites (Figure 1d,e). 

The patient’s first-degree relatives underwent genetic screening for the same RET variant (Figure 1f). This showed heterozygosity in the patient’s father and daughter, but their phenotypes did not show any pathological alterations. Past medical history, as well as hormonal and imaging assessments, were unremarkable.

At the last follow-up in June 2023, the patient was disease-free for breast cancer, without other neoplasms. Acromegaly disease was biochemically controlled after five years of treatment with Pasireotide Lar 60 mg/monthly in association with Pegvisomant 20 mg/daily and Cabergoline 1 mg/daily, with a stable tumor remnant. The patient reported that her 25-year-old daughter underwent a left quadrantectomy for a breast fibro-adenoma.

## 3. Discussion

RET (10q11.2) is a proto-oncogene encoding for a receptor tyrosine kinase protein. The first clinical manifestation of the RET mutation was identified in papillary thyroid carcinomas (PTCs) in 1990, where genomic rearrangements led to chimeric RET/PTCs oncoproteins [6]. Loss-of-function RET mutations were reported to be associated with Hirschsprung disease, while gain-of-function RET mutations were reported to be associated with MEN2A, MEN2B, CCHS, pheochromocytoma, renal agenesis, and HCPS [7]. In several tumor cell types, the activation of RET signaling proved to promote tumor proliferation and progression [8]. Somatic RET oncogenic mutations are also found in 50% of sporadic medullary thyroid carcinomas (SMTC). RET amplifications were reported in several cancers located in the thyroid, pancreas, and breast [6,22].

Germline RET mutations generally occur in the context of specific syndromes, such as MEN2. Somatic RET mutations tend to be associated with certain types of tumors, such as breast cancer or SMTC [9]. The c.2410G>A PV identified in our patient is the same as reported in MEN2A, MEN2B, FMTC, pheochromocytoma, HCPS, renal hypodysplasia/aplasia, CCHS, and Hirschsprung disease [10]. As with most germline RET point mutations, this alteration was heterozygous. As recently reported by Schirwani et al. [8], there are limited reports of individuals with homozygosity for high-risk pathogenic RET variants, particularly for the low penetrance variants p.Val804Met and p.Val804Leu [12,13]. In a large consanguineous family segregating the p.Val804Met variant, MTC only developed in the four homozygous individuals, while all heterozygotes had normal screening results [12]. In a retrospective study investigating a total of 306 cases, three homozygous patients with either p.Val804Leu or p.Val804Met were identified. Their disease features were similar to their heterozygous counterparts, and they presented no known family history of MEN2 [13]. No cases of acromegaly alone were reported in association with the Val804Met alteration. Nevertheless, as reported above (and considering the huge amount of data on the mutational effect given on each affected RET-domain [11]), we could still definitively make the assessment that the V804M alteration is a low penetrance mutation. The related phenotype can be due to both environmental and possibly coexisting modifier genes. Unfortunately, data regarding these modifiers are not yet available in the current literature.

The RET gene mutations and fusions are the drivers of many cancer types (Table 1). Mutations in the extracellular cysteine-rich domain led to constitutive RET receptor dimerization and activation, regardless of their ligands. As well, those falling in the tyrosine kinase domains cause constitutive kinase activation. *RET* rearrangements produce chimeras determined by the fusion of an N-terminal partner to the RET C-terminal portion, leading to aberrant *RET* overexpression, kinase activation, and ligand-independent dimerization [9,23]. The presence of RET fusions is well documented in breast cancer, but *RET* germline PVs were never reported in breast cancer. On the contrary, the *RET* PV carried by our patient was indeed reported in MEN2 and the other conditions mentioned above, including HCPS. 

Furthermore, we considered the possibility that the RET pathogenic variant may be a contributing factor in the susceptibility to breast tumors. This was due to the following observations: the diagnosis of breast tumors in both the female individuals that we are reporting, the early onset of breast fibro-adenoma in the daughter of our patient, and the absence of other variants strongly associated with familial cancers. We may also speculate that it could be a hereditary predisposition to develop tumors under these manifestations, framing in the HCPS, a known *RET* PV (p.Val804Met)-related syndrome. 

The *RET* protooncogene is widely expressed in the pituitary [10]. RET and its co-receptor GFRa1 are expressed exclusively in somatotroph cells. RET is implicated in maintaining the normal number of somatotroph cells and the physiological GH levels, promoting cell survival, and down-regulating GH production [15]. In our patient, the *RET* mutation could have induced constitutive activation of tyrosine kinase (also expressed in the pituitary), enhancing GH production and somatotroph cell proliferation, leading to invasive growth and a poor response to treatments. 

Interestingly, our patient did not develop MTC or other MEN-2 syndrome-related neoplasia. The presentation of MEN-2A coincides rarely with PitNET. Heinlen et al. demonstrated a *RET* missense mutation in a MEN-2A-affected patient, presenting a non-secreting PitNET [16]. To our knowledge, only two cases of acromegaly have been associated with *RET* PVs in the context of MEN-2A [17,18]. 

Moreover, the penetrance of *RET* mutations can determine different phenotypes. It was recently reported that there are gender differences in MEN-2A penetrance and expression according to parental inheritance. This suggests a different risk of tumor onset and could determine different phenotypes in MEN2A patients [19]. Finally, different PVs can cause different levels of receptor activation, and functional impact tends to correlate with clinical presentation [20,24]. Given these considerations, our clinical family histories support the hypothesis that *RET* c.2410G>A PV caused the HCPS [25,26]. HCPS is an inherited disorder with an increased cancer risk. HCPS is caused by germline RET mutations and is characterized by an early age of tumor onset, the presence of two or more different neoplasias in the same patient, and the occurrence of the same type of neoplasia within a family [27,28,29]. As summarized in Table 2, over 50 patients/families affected by HCPS have been described. 

## 4. Conclusions

Our clinical case described for the first time a *RET* PV (p.Val804Met) in acromegaly in the absence of other known phenotypic manifestations of the *RET* mutation. Our findings suggest that we should screen patients with aggressive tumors and suggestive clinical histories for *RET* mutations in cases with NGS for well-established PitNET-associated gene mutations rendered negative. We advocate for further studies to corroborate the relationship between *RET* pathogenic variants and acromegaly.

## Figures and Tables

**Figure 1 ijms-25-01895-f001:**
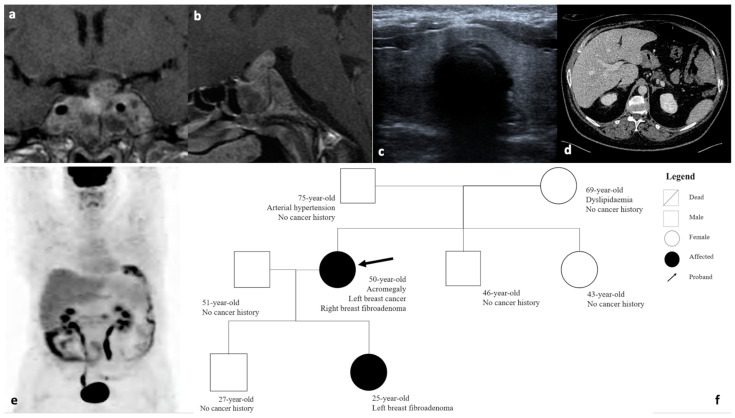
Radiological and family history of a *RET*-mutated acromegaly patient. Coronal (**a**,**b**) sagittal pituitary T1-weighted MRI showed a giant pituitary tumor invading both cavernous sinuses and the clivus. Thyroid ultrasound (**c**) showed a right lobe thyroid nodule. The abdominal CT (**d**) ruled out the presence of adrenal nodules and pheochromocytoma. (**e**) The 18F-fluorodeoxyglucose positron emission tomography/computed tomography (18F-FDG PET/TC) excluded the presence of breast cancer metastasis and other neoplasia. (**f**) The family’s genealogical tree documented that the patient’s mother, brother, sister, and son had no cancer history; the proband’s father and daughter carried the same germline RET pathogenic variant.

**Table 1 ijms-25-01895-t001:** Molecular mechanisms of RET activation according to mutation and fusion and associated neoplasia.

Molecular Mechanisms of RET Activation			
*RET* Mutations		*RET* Fusions	
Ligand Independent Dimerization	Ligand Independent Activation	Aberrant RET Expression	Constitutive RET Activation
▪ Multiple Endocrine Neoplasia type 2 (MEN2) ▪ Medullary thyroid carcinoma (MTC) ▪ Pheochromocytoma ▪ Congenital central hypoventilation syndrome (CCHS) ▪ Hirschsprung disease ▪ Renal agenesis ▪ Hereditary cancer-predisposing syndrome (HCPS)		▪ Papillary thyroid carcinoma (PTC) ▪ Poorly differentiated thyroid cancer (PDTC) ▪ Non-small cell lung cancer (NSCLC) ▪ Small cell lung cancer (SMLC) ▪ Breast cancer ▪ Prostate cancer ▪ Pancreatic cancer ▪ Leukemia ▪ Colon–rectal cancer ▪ Melanoma ▪ Salivary gland adenocarcinoma ▪ Esophago-gastric cancer ▪ Intrahepatic cholangiocarcinoma ▪ Hepatobiliary cancer ▪ Liver hepatocellular carcinoma ▪ Ovarian epithelial tumor ▪ Spitz tumor	

**Table 2 ijms-25-01895-t002:** Summary of neoplasia reported in hereditary cancer-predisposing syndrome.

Hereditary Predisposing Cancer Syndromes	
**Skin neoplasia**	Gorlin syndrome or nevoid basal cell carcinoma syndrome Bloom syndrome Epidermolysis bullosa Rothmund–Thomson syndrome Werner syndrome Xeroderma pigmentosum Brooke–Spiegler syndrome Familial cylindromatosis Cowden syndrome and *PTEN* Hamartoma Tumor syndromes Fanconi Anemia Multiple familial trichoepithelioma Oculocutaneous albinism Melanoma, hereditary Muir–Torre syndrome Epidermodysplasia verruciformis Dyskeratosis congenita (Zinsser–Cole–Engman Syndrome)
**Breast and gynecologic neoplasia**	Hereditary breast/gynecologic cancers Cowden syndrome and *PTEN* Hamartoma tumor syndromes Li–Fraumeni syndrome Gastric cancer with diffuse and lobular breast cancer
**Endocrine neoplasia**	Carney–Stratakis syndrome Familial hyperparathyroidism Hereditary paraganglioma and hereditary pheochromocytoma Familial medullary thyroid cancer Multiple endocrine neoplasia type 1 (Wermer syndrome) Multiple endocrine neoplasia type 2A and 2B (Sipple syndrome)
**Colon-rectal neoplasia**	Hereditary nonpolyposis colon cancer or Lynch syndrome Cowden syndrome and *PTEN* Hamartoma tumor syndromes Fanconi anemia Oligopolyposis Peutz–Jeghers syndrome Familial adenomatous polyposis and attenuated familial Adenomatous polyposis Familial juvenile polyposis Hereditary mixed polyposis *MUTYH*-associated polyposis
**Others**	Von Hippel–Lindau disease Birt–Hogg–Dubé syndrome Prostate Cancer, hereditary Hereditary renal cell cancer with uterine leiomyomas Hereditary papillary Renal cell cancer

## Data Availability

The authors confirm that the findings of this study are available within the article.

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
