# Peer review of "The Pathogenic RET Val804Met Variant in Acromegaly: A New Clinical Phenotype?"

_ijms, 2024, doi:10.3390/ijms25031895_

Round 1

Reviewer 1 Report

Comments and Suggestions for Authors

Reviewer comments on manuscript # ijms-2801897, entitled “The pathogenic RET Val804Met variant in acromegaly: a new clinical phenotype?”,

Comment #1, all manuscript: the authors should review the English language of the manuscript, preferably with a native English speaker, as there are several language errors throughout the text. Some (but not all) examples are:

Line 33 – “…contest of somatotropinomas and other endocrine tumors …” context not contest.

 Lines 46-47 - “..with an unpredictable and rapid evolution affecting the patients’ overall Genetic discoveries..” ?? Lack of words to terminate the sentence? Lack of comma before “Genetic”.

Lines 56-57 – “”..such as AIP, MEN1, CDKN1B, GPR101, PRKAR1A, and GNAS,4 in turn impacting the therapeutic approach and prognosis.” besides the lack of italic style when describing genes, the phrase is too confuse and the number four is probably a mistake..

Line 60 – “AIP-mutated patients typically carried isolated and invasive pituitary macroadenoma…” carried is not the word. Consider instead develop/present/manifest..  

Line 102 – “..fg-SRLs..” the authors shoud describe all the abbreviations. Thus, they should proceed in accordance to “fg” too.

Line 142 – “..with stable tumor residual,..” consider remnant instead.

Lines 163-164 – “gain-of-function mutations for MEN2A, MEN2B, CCHS, pheochromocytoma, renal agenesis, and HCPS” the sentence is miswritten and confuse..

Line 173 – “..this alteration was in heterozygosis.” “Heterozygosis” is unusual. Please consider instead heterozygosity.

Lines 111 and 211 – the authors use the word PitNET to refer to pituitary adenomas. However, in the abovementioned lines of the manuscript they use PT but never discriminate this abbreviation, which is not consistent with the abbreviation PitNET…please revise.

Comment #2, lines 65-66: “MEN1 mutations have a prevalence of 1-2% PitNETs”, besides miswritten, there is doubt if the authors wanted to say penetrance (and thus it is much more than 2%) or prevalence of MEN1 mutations in patients with apparently sporadic PitNets.  

Comment #3, lines 73 and 75: “somatolactotropinoma” is an unusual word. As the authors wrote in previous lines, mammosomatotropinoma or somatomammotropinoma are most common used word for this histologic subtype of PitNET. 

Comment #4, lines 86-87: “Nowadays, RET mutations have not been reported in acromegaly.” This sentence is not true. Please refer to the paper of Saito T, et al in Am. J. Med. Sci. 2010340, 329–331. Please revise this and all the other sentences in the text where you refer this case of acromegaly as the first occurring in a patient with a RET mutation.

Comment #5, lines 131-132: and cytological examinations excluded thyroid malignant neoplasia”. Why the patient performed FNA? The authors state in Figure 1 that “Thyroid ultrasound (1c) showed a normal gland structure without nodules.”….please clarify/revise. 

Comment #6, lines 165-166: “RET oncogenic mutations are also found in 50% of sporadic medullary thyroid carcinomas” the authors want to refer to somatic mutations? Please clarify/revise.

Comment #7, Lines 36-37 and 246: “…and suggests that RET may be included in the gene panel of aggressive somatotropinomas..” AND “RET mutations may be included in the gene panel of somatotropinomas, in patients with aggressive tumors and suggestive clinical history, particularly in the context of clinical exome screenings.” The relationship between RET mutations and PitNETs remains debatable and the word coincident is usually used to describe this association. This is due to the fact that only three case reports were published to date (patients with RET mutations and PitNETs) and no RET mutations were reported in sporadic or familial PitNET cases. Additionally, the penetrance of the prototypic phenotype of patients with the p.Val804Met RET variant is ~4% (J Clin Endocrinol Metab. 2018; 103: 4275–4282), which leads to the conclusion that this variant has probably a low/moderate oncogenic risk. Thus, the authors suggestion to include this gene in NGS constructs for PitNETs seems futile. I would recommend the authors to advise the reader to consider screening for RET mutations if NGS for well-established PitNETs-associated gene mutations renders negative.

Comments on the Quality of English Language

Moderate improvements needed..

Author Response

Comment #1, all manuscript: the authors should review the English language of the manuscript, preferably with a native English speaker, as there are several language errors throughout the text. Some (but not all) examples are:

Line 33 – “…contest of somatotropinomas and other endocrine tumors …” context not contest.

We thank the reviewer for the comment. Our paper has been revisioned by native English speaker. We modified at line 32.

 Lines 46-47 - “..with an unpredictable and rapid evolution affecting the patients’ overall Genetic discoveries..” ?? Lack of words to terminate the sentence? Lack of comma before “Genetic”.

We thank the reviewer for the comment. We modified at line 41.

Lines 56-57 – “”..such as AIP, MEN1, CDKN1B, GPR101, PRKAR1A, and GNAS,4 in turn impacting the therapeutic approach and prognosis.” besides the lack of italic style when describing genes, the phrase is too confuse and the number four is probably a mistake..

We thank the reviewer for the comment. We modified at line 51.

Line 60 – “AIP-mutated patients typically carried isolated and invasive pituitary macroadenoma…” carried is not the word. Consider instead develop/present/manifest..  

We thank the reviewer for the comment. We modified at line 58.

Line 102 – “..fg-SRLs..” the authors shoud describe all the abbreviations. Thus, they should proceed in accordance to “fg” too.

We thank the reviewer for the comment. We modified at line 100.

Line 142 – “..with stable tumor residual,..” consider remnant instead.

We thank the reviewer for the comment. We modified at line 143.

Lines 163-164 – “gain-of-function mutations for MEN2A, MEN2B, CCHS, pheochromocytoma, renal agenesis, and HCPS” the sentence is miswritten and confuse.

We thank the reviewer for the comment. We modified at line 161.

Line 173 – “..this alteration was in heterozygosis.” “Heterozygosis” is unusual. Please consider instead heterozygosity.

We thank the reviewer for the comment. We modified at line 173.

Lines 111 and 211 – the authors use the word PitNET to refer to pituitary adenomas. However, in the above mentioned lines of the manuscript they use PT but never discriminate this abbreviation, which is not consistent with the abbreviation PitNET…please revise.

We thank the reviewer for the comment. We modified at line 110 and 210.

Comment #2, lines 65-66: “MEN1 mutations have a prevalence of 1-2% PitNETs”, besides miswritten, there is doubt if the authors wanted to say penetrance (and thus it is much more than 2%) or prevalence of MEN1 mutations in patients with apparently sporadic PitNets.  

We thank the reviewer for the comment. We clarified and modified, reporting the exact percentage (0.6–2.6%) for MEN1 mutations prevalence in pituitary tumors (BogusÅ‚awska A, Korbonits M. Genetics of Acromegaly and Gigantism. J Clin Med. 2021 Mar 29;10(7):1377. doi: 10.3390/jcm10071377. PMID: 33805450; PMCID: PMC8036715) at line 63.

Comment #3, lines 73 and 75: “somatolactotropinoma” is an unusual word. As the authors wrote in previous lines, mammosomatotropinoma or somatomammotropinoma are most common used word for this histologic subtype of PitNET. 

We thank the reviewer for the comment. We modified at line 71 and 73.

Comment #4, lines 86-87: “Nowadays, RET mutations have not been reported in acromegaly.” This sentence is not true. Please refer to the paper of Saito T, et al in Am. J. Med. Sci. 2010, 340, 329–331. Please revise this and all the other sentences in the text where you refer this case of acromegaly as the first occurring in a patient with a RET mutation.

We thank the reviewer for the comment, and for highlighting one of the two cases we identified and reported at line 210.

As we wrote, to our knowledge, only two cases of acromegaly have been associated with RET PVs in the context of MEN-2A [Saito T, Miura D, Taguchi M, Takeshita A, Miyakawa M, Takeuchi Y. Coincidence of multiple endocrine neoplasia type 2A with acromegaly. Am J Med Sci. 2010;340(4):329-31. doi: 10.1097/MAJ.0b013e3181e73fba. PMID: 20739875. /// Machens A, Lorenz K, Weber F, Dralle H. Sex differences in MEN 2A penetrance and expression according to parental in-heritance. Eur J Endocrinol. 2022;186(4):469-476. doi: 10.1530/EJE-21-1086. PMID: 35130180.].

We thank the reviewer for giving us the opportunity to underline the extraordinary nature of this case of acromegaly as the first occurring in a patient with a RET pathogenic variant not in the context of a MEN. We modified at line 85.

Comment #5, lines 131-132: “and cytological examinations excluded thyroid malignant neoplasia”. Why the patient performed FNA? The authors state in Figure 1 that “Thyroid ultrasound (1c) showed a normal gland structure without nodules.”….please clarify/revise. 

We thank the reviewer for the comment. We apologize and modified at line 150.

Comment #6, lines 165-166: “RET oncogenic mutations are also found in 50% of sporadic medullary thyroid carcinomas” the authors want to refer to somatic mutations? Please clarify/revise.

We thank the reviewer for the comment. We clarified that we refer to somatic mutations and we detailed at page 4 line 166.

Comment #7, Lines 36-37 and 246: “…and suggests that RET may be included in the gene panel of aggressive somatotropinomas..” AND “RET mutations may be included in the gene panel of somatotropinomas, in patients with aggressive tumors and suggestive clinical history, particularly in the context of clinical exome screenings.”

The relationship between RET mutations and PitNETs remains debatable and the word coincident is usually used to describe this association. This is due to the fact that only three case reports were published to date (patients with RET mutations and PitNETs) and no RET mutations were reported in sporadic or familial PitNET cases. Additionally, the penetrance of the prototypic phenotype of patients with the p.Val804Met RET variant is ~4% (J Clin Endocrinol Metab. 2018; 103: 4275–4282), which leads to the conclusion that this variant has probably a low/moderate oncogenic risk. Thus, the authors suggestion to include this gene in NGS constructs for PitNETs seems futile. I would recommend the authors to advise the reader to consider screening for RET mutations if NGS for well-established PitNETs-associated gene mutations renders negative.

We thank the reviewer for the comment. We modified and added your important suggestion at line 35 and 247.

Reviewer 2 Report

Comments and Suggestions for Authors

The presented study investigates the genetic landscape of somatotropinomas, focusing on the identification of the RET pathogenic variant, c.2410G>A (rs79658334), p.Val804Met, in an acromegaly patient—a unique finding not reported before. The familial occurrence of this variant in the father and daughter of the patient adds an intriguing dimension. The study explores the clinical implications of this variant in the context of somatotropinomas and other endocrine tumors, exploring the oncogenic mechanisms of RET mutations for potential precision interventions in the treatment of acromegaly. However, some minor adjustments are needed to enhance clarity and coherence before the manuscript can be considered for publication.

Introduction

I would like to suggest a refinement in the Introduction section. Currently, the sentences discussing different genetic changes appear somewhat disjoint. To enhance the coherence and clarity of your work, I kindly recommend restructuring these sentences into one or two well-organized paragraphs. This adjustment will not only improve the overall flow of the introduction but also provide readers with a more comprehensive understanding of the genetic alterations.

Case Description

 Page 03 – lines 110 – 115 – “According to the somatotropinoma’s aggressive behavior, and patient’s cancer history, by clinical exome sequencing (CES) a virtual panel was designed for PT, encompassing the genes: ABCC, ABCC8, AIP, APC, ATP2B3, CACNA1D, CACNA1H, CDKN1B, CLCN2, DICER1, DIS3L2, FGFR3, GCK, GIPR, GNAS, HNF1A, HNF4A, HRAS, IGF1R, KCNJ5, LHX3, MEN1, NF1, NSD1, PDE11A, PDE8B, POU1F, PRKAR1A, PROP1, PTCH1, PTEN, RET, SDHA, SDHB, SDHC, SDHD, SHANK3, SLC16A1, SLC34A1, TSC1, TSC2, VHL.”   

The provided text lacks sufficient information for the reader, and a more informative approach could be achieved by incorporating a supplementary table that details the genes and their respective functions. Additionally, it would be beneficial to include a reference explaining the rationale behind selecting these specific genes for clinical exome sequencing in the context of somatotropinoma's aggressive behavior and the patient's cancer history.

Page 03 – lines 120 – 129 – “After an accurate variant calling process, the only pathogenic variant reported by the SoPHiA DDM software was the c.2410G>A (rs79658334), p.Val804Met variant in the RET gene. We further extended the analysis to the other genes outside the virtual panel, and no pathogenic variants were identified. To clarify the meaning of this mutation and its function, the genetic research portals ClinVar, Varsome, Franklin Uniprot, Ensembl, OMIM, Gene Cards, gnomAD were consulted and, unanimously, this variant was classified as pathogenic (Class 5). These website portals reported this pathogenic variant (PV) as associated to MEN-2A and MEN-2B, familial medullary thyroid carcinoma (FMTC), pheochromocytoma, hereditary cancer-predisposing syndrome (HCPS), renal hypodysplasia or aplasia, congenital central hypoventilation syndrome (CCHS), and Hirschsprung disease.”

The authors should incorporate references in the text detailing the methodology employed by the SoPHiA DDM software for identifying pathogenic variants. It is recommended that the authors adhere to established guidelines for variant classification, such as those outlined in the work by Richards et al. (doi: 10.1038/gim.2015.30.).

Author Response

The presented study investigates the genetic landscape of somatotropinomas, focusing on the identification of the RET pathogenic variant, c.2410G>A (rs79658334), p.Val804Met, in an acromegaly patient—a unique finding not reported before. The familial occurrence of this variant in the father and daughter of the patient adds an intriguing dimension. The study explores the clinical implications of this variant in the context of somatotropinomas and other endocrine tumors, exploring the oncogenic mechanisms of RET mutations for potential precision interventions in the treatment of acromegaly. However, some minor adjustments are needed to enhance clarity and coherence before the manuscript can be considered for publication.

Introduction

I would like to suggest a refinement in the Introduction section. Currently, the sentences discussing different genetic changes appear somewhat disjoint. To enhance the coherence and clarity of your work, I kindly recommend restructuring these sentences into one or two well-organized paragraphs. This adjustment will not only improve the overall flow of the introduction but also provide readers with a more comprehensive understanding of the genetic alterations.

We thank the reviewer for the comment. Our paper has been revisioned by native English speaker. We modified.

Case Description

 Page 03 – lines 110 – 115 – “According to the somatotropinoma’s aggressive behavior, and patient’s cancer history, by clinical exome sequencing (CES) a virtual panel was designed for PT, encompassing the genes: ABCC, ABCC8, AIP, APC, ATP2B3, CACNA1D, CACNA1H, CDKN1B, CLCN2, DICER1, DIS3L2, FGFR3, GCK, GIPR, GNAS, HNF1A, HNF4A, HRAS, IGF1R, KCNJ5, LHX3, MEN1, NF1, NSD1, PDE11A, PDE8B, POU1F, PRKAR1A, PROP1, PTCH1, PTEN, RET, SDHA, SDHB, SDHC, SDHD, SHANK3, SLC16A1, SLC34A1, TSC1, TSC2, VHL.”   

The provided text lacks sufficient information for the reader, and a more informative approach could be achieved by incorporating a supplementary table that details the genes and their respective functions. Additionally, it would be beneficial to include a reference explaining the rationale behind selecting these specific genes for clinical exome sequencing in the context of somatotropinoma's aggressive behavior and the patient's cancer history.

Regarding the question of gene panel selected by Exome sequencing bioinformatic analysis, we do not agree with the reviewer regarding the need to specify the role of every gene included in the list of clinical exome. Scientists running WES in routine or translational settings can filter gene by disease or syndrome using the appropriate software associated to the NGS pipeline. In this case, we used the SOPHiA DDM software that is based on overall information linking the specific syndrome or disease to a panel of genes (rarely single gene) that have been published in literature as associated to germline or somatic alterations. However, since the layout of SOPHiA DDM is covered by patent, we cannot provide any information about the reason why, when we inputted the query “Pituitary tumors”, the software selected only the genes listed in our paper. Nevertheless, genes reported in our pipeline are the same listed in the papers by EC Coopmans et al. (Molecular genetic testing in the management of pituitary disease;  Clin Endocrinol, 97 (2022), p. 424), by  Daniel Marrero-Rodríguez et al. In Arch Med Res . 2023 Dec;54(8):102915, and by J Denes et al (The clinical aspects of pituitary tumors genetics) in Endocrine, 71 (2021), as associated to PTs.

The aggressive behavior is the phenotype describing patients’ status in relationship to the incidental gene alteration. We have explained that there are many mechanisms determining the individual phenotype in PTs that are also heterogenous syndromes.

Page 03 – lines 120 – 129 – “After an accurate variant calling process, the only pathogenic variant reported by the SoPHiA DDM software was the c.2410G>A (rs79658334), p.Val804Met variant in the RET gene. We further extended the analysis to the other genes outside the virtual panel, and no pathogenic variants were identified. To clarify the meaning of this mutation and its function, the genetic research portals ClinVar, Varsome, Franklin Uniprot, Ensembl, OMIM, Gene Cards, gnomAD were consulted and, unanimously, this variant was classified as pathogenic (Class 5). These website portals reported this pathogenic variant (PV) as associated to MEN-2A and MEN-2B, familial medullary thyroid carcinoma (FMTC), pheochromocytoma, hereditary cancer-predisposing syndrome (HCPS), renal hypodysplasia or aplasia, congenital central hypoventilation syndrome (CCHS), and Hirschsprung disease.”

The authors should incorporate references in the text detailing the methodology employed by the SoPHiA DDM software for identifying pathogenic variants. It is recommended that the authors adhere to established guidelines for variant classification, such as those outlined in the work by Richards et al. (doi: 10.1038/gim.2015.30.).

The classification of the alteration is based on the ClinVAr database, where for the variant NM_020975.4(RET):c.[2410G>A] is reported as Pathogenic following the (ACMG Guidelines, 2015). So we referred to the most used clinical genomic database where the variant are classified following the international  guidelines ACGM2015.

We cannot establish a priori the biological effect of this pathogenic variant in overall individual body cells, but we have only reported the association between genotype and phenotype. 

Reviewer 3 Report

Comments and Suggestions for Authors

Please see the attached review.

Comments on the Quality of English Language

Extensive editing of English language required

Author Response

Thank you for the opportunity to review the manuscript “The pathogenic RET Val804Met variant in acromegaly: a new clinical phenotype?” by Sabrina Chiloiro et al. Acromegaly is a rare disease and it is valuable to study mechanisms of its possible genetic origin. However, there are some issues that need to be clarified and improved to consider the paper for publication. I would recommend to revise the paper thoroughly and resubmit.

The style and grammar makes it difficult to understand the text, e.g.:

Line 29- Somatotropinomas are the pituitary tumor subtypes more genetically investigated for germline and somatic mutations.

We thank the reviewer for the comment. We rephase at line 28.

Lines 35-36- Our case describes a new phenotype associated with a RET pathogenic variant, represented by aggressive acromegaly, and suggests that RET may be included in the gene panel of aggressive somatotropinomas, particularly in the context of clinical exome screenings.

We thank the reviewer for the comment. We modified this sentence according to the indications of Reviewer 2.

Line 58- 29% in acro-gigantism

We thank the reviewer for the comment. We clarified and modified at line 56, reporting the exact percentage from table1 of BogusÅ‚awska A, Korbonits M. Genetics of Acromegaly and Gigantism. J Clin Med. 2021 Mar 29;10(7):1377. doi: 10.3390/jcm10071377. PMID: 33805450; PMCID: PMC8036715.   

Line 60- AIP-mutated patients Introduction should be more comprehensive not just cites all the genes. Genes names should be written in italic and the abbreviations should be explained.

We thank the reviewer for the comment. We added at line 51.

Line 49- insulin-like growth factor 1 (IGF-I) should be discussed separately as not a pituitary hormone.

 We thank the reviewer for the comment. We modified at line 47.

Figure 1 and Table 1, 2 are unclear.

We thank the reviewer for the comment. We clarified.

Conclusions- rather like Discussion

References- not sufficient

We thank the reviewer for the comment. We added some references.

Round 2

Reviewer 3 Report

Comments and Suggestions for Authors

Please see the attached review.

Comments on the Quality of English Language

Minor editing of English language required

Author Response

We thank the reviewer for the suggestions and comments, that we followed, allowing us to improve the quality of our paper. We reviewed the results and the conclusion. A native English speaker reviewed our manuscript.